# Protective Effects of Bromelain in Testicular Torsion-Detorsion: Reducing Inflammation, Oxidative Stress, and Apoptosis While Enhancing Sperm Quality

**DOI:** 10.3390/biom15020292

**Published:** 2025-02-15

**Authors:** Seda Yakut, Merve Karabulut, Recep Hakkı Koca, Elif Erbaş, Seçkin Özkanlar, Berrin Tarakçı Gençer, Adem Kara, K. J. Senthil Kumar

**Affiliations:** 1Department of Histology and Embryology, Faculty of Veterinary Medicine, Burdur Mehmet Akif Ersoy University, Burdur 15030, Türkiye; syakut@mehmetakif.edu.tr; 2Department of Surgery, Faculty of Veterinary Medicine, Bingöl University, Bingöl 12000, Türkiye; mkarabulut@bingol.edu.tr; 3Department of Reproduction and Artificial Insemination, Faculty of Veterinary Medicine, Bingöl University, Bingöl 12000, Türkiye; rhkoca@bingol.edu.tr; 4Department of Histology and Embryology, Faculty of Medicine, Atatürk University, Erzurum 25240, Türkiye; eliferbas@atauni.edu.tr; 5Department of Biochemistry, Faculty of Veterinary Medicine, Atatürk University, Erzurum 25240, Türkiye; seckinozkanlar@yahoo.com; 6Department of Histology and Embryology, Faculty of Veterinary Medicine, Fırat University, Elazığ 23119, Türkiye; btarakci@firat.edu.tr; 7Department of Molecular Biology and Genetics, Faculty of Science, Erzurum Technical University, Erzurum 25100, Türkiye; adem.kara@erzurum.edu.tr; 8Department of Biotechnology, Saveetha Institute of Medical and Technical Sciences, Saveetha University, Thandalam, Chennai 602105, Tamil Nadu, India; 9Center for General Education, National Chung Hsing University, Taichung 402, Taiwan

**Keywords:** Bromelain, testicular torsion, ischemia/reperfusion injury, NRF-2/HO-1, PI3K/Akt/mTOR, apoptosis

## Abstract

Inflammation and increased oxidative stress in testicular tissue are documented side effects of torsion of the testicles. The preventive role of Bromelain (Bro) against testicle torsion-induced ischemia/reperfusion (I/R) injury was investigated in this research. Five groups of six animals each were created: ischemia, Ischemia+Reperfusion (I+R), Ischemia+Reperfusion+Bromelain (I+R+Bro; 10 mg/kg), control (sham), and Bromelain (Bro; 10 mg/kg). An I/R damage resulted from two hours of 720° clockwise twisting of the left testis. Blood samples and epididymal sperm were collected after reperfusion to analyze sperm parameters (recovery, motility, viability, and morphology) and cytokines that promote inflammation (IL-1β, IL-6, and TNF-α). Using Western blotting, testicular tissue was examined for histopathological alterations, antioxidant enzymes (GSH, SOD), lipid peroxidation (MDA), apoptosis, and survival-related proteins (TLR4, Caspase-3, Bcl-2, NRF-2, HO-1, PI3K, mTOR, AKT-1). While raising the activities of GSH and SOD, two antioxidant enzymes, Bro administration dramatically reduced MDA concentrations. The I+R+Bro group had significantly reduced amounts of cytokines that promoted inflammation compared to the I+R group. Bro’s protective properties are also attributed to proteins that are altered by it and participate in the apoptosis and survival of cells. Sperm morphology, motility, and concentration notably improved in the bromelain-treated group, according to spermatological examination. Testicular samples treated with bromelain showed less tissue damage according to histological evaluations than the untreated I+R group. These findings imply that Bro has anti-inflammatory, anti-apoptotic, and antioxidant qualities. It effectively reduces oxidative stress and inflammation by modulating the PI3K/Akt/mTOR and NRF-2/HO-1 pathways, hence minimizing I/R injury.

## 1. Introduction

Testicular torsion, where the testicle twists around its axis, resulting in ischemia and reperfusion (I/R) injury, is an urgent urological condition requiring prompt surgical treatment to avoid tissue damage and loss [1]. One of the main causes of testicular ischemia is testicular torsion, which results in inflammation, the loss of germ cells, and oxidative stress [2]. This disorder lowers antioxidant capacity and encourages the generation of reactive oxygen species (ROS), the buildup of calcium in the mitochondria, and apoptosis, all of which lead to ischemic damage [3,4,5]. Various studies have demonstrated that treatment with antioxidant agents is a viable option to lessen the impact of damage caused by torsion or detorsion [6,7]. Compounds such as Baicalein and Proanthocyanidin have been found to exhibit protective effects by controlling oxidative stress and reducing inflammation [8,9].

It has been demonstrated that the pineapple-derived protease enzyme bromelain (Bro) exhibits antioxidant action by raising the antioxidant capacity and decreasing oxidative stress indicators such as lipid peroxidation [1,10]. The neuroprotective [11] and anticarcinogenic [12] properties of Bro have been subject to investigation, revealing its potential to attenuate the apoptotic cascade across diverse cancer types and neurodegenerative disorders. Supplementation with bro has been demonstrated to ameliorate sperm count, enhance normal sperm morphology, elevate levels of testosterone, regulate expression of estrogen receptors, and augment mice’s antioxidant enzyme activity subjected to Bisphenol A(BPA) exposure. Bro demonstrated a protective role by reversing the effects of BPA, which enhanced oxidative stress by increasing MDA and decreasing antioxidant enzyme activity [13].

Using an experimental model of torsion-induced I/R injury, this research investigates the effects of Bro on testicular tissue. A key area of study is the signaling pathway involving phosphoinositide 3-kinase (PI3K), protein kinase B (Akt), and the mammalian target of rapamycin (mTOR). This system is essential for male reproduction and regulates the hypothalamic–pituitary–gonadal (HPG) axis during spermatogenesis. Our study also targets the NRF-2/HO-1 pathway, which is connected to immune-regulatory and anti-inflammatory mechanisms, as well as proteins linked to apoptosis, such as Bcl-2 and Caspase-3. Additionally, this study assesses cytokines that promote inflammation (IL-6, IL-1β, and TNF-α), indicators of oxidative stress, and Toll-like receptor 4 (TLR4), a crucial element of the immune response.

## 2. Materials and Methods

### 2.1. Chemicals

Sigma-Aldrich Chemicals provided all reagents and compounds, including Bromelain (CAS No: 37189-34-7), which were of analytical degree.

### 2.2. Animals and Experimental Design

Thirty male Wistar albino rats, weighing between 200 and 250 g and aged 8 to 12 weeks, participated in this investigation. The rats were obtained from the Experimental Research and Application Center of Burdur Mehmet Akif Ersoy University (Burdur, Turkey). The rats were housed in cages under a 12 h light/dark cycle (light: 06:00–18:00; dark: 18:00–06:00) at a controlled temperature of 25 °C. They received regular chow pellets and were given unlimited access to food and water. The Burdur Mehmet Akif Ersoy University Local Ethics Committee gave its approval for all experimental protocols (Approval No. 1293). The thirty adult male rats were randomly assigned to five groups, with each group comprising six rats (n = 30), ensuring similar body weight distribution across groups. Eraky et al.’s earlier research served as the basis for the dosage of Bro utilized in this investigation [14]. The surgical procedure was established according to the procedure used by Wei and Huang [8]. The study groups and their respective treatments are detailed as follows: Control Group (Sham): For baseline measurements, an incision was made in the scrotal area and closed with 4/0 sutures. PBS was administered intraperitoneally thirty minutes before the orchiectomy procedure (i.p.). Two hours later, the sutures were removed, and the orchiectomy was performed. Bromelain Group (Bro): For baseline measurements, an incision was made in the scrotal area and closed with 4/0 sutures. Thirty minutes before the orchiectomy procedure, a dose of 10 mg/kg of Bro was given intraperitoneally (i.p.). In order to concentrate on its preventive rather than therapeutic effect, Bro was applied before orchiectomy. Two hours later, the sutures were removed, and orchiectomy was performed. Ischemia Group: Left testes underwent 720° counterclockwise torsion for 2 h. PBS was administered intraperitoneally thirty minutes before the orchiectomy procedure (i.p.). Two hours later, the sutures were removed, and orchiectomy was performed. Ischemia + Reperfusion Group (I+R): Left testes underwent 720° counterclockwise torsion. After 2 h of ischemia, the testes were detorsed. Orchiectomy was performed 2 h after reperfusion. PBS was administered intraperitoneally thirty minutes before the orchiectomy procedure (i.p.). Ischemia + Reperfusion + Bromelain Group (I+R+Bro): The left testes underwent 720° counterclockwise torsion. After 2 h of ischemia, the testes were detorsed. Orchiectomy was performed 2 h after reperfusion. Thirty minutes before the orchiectomy procedure, a dose of 10 mg/kg of Bro was given intraperitoneally (i.p.). At the culmination of the research study, the collection of samples marked the end phase. Blood samples were gathered via intracardiac extraction from all subjects, followed by euthanasia via decapitation. These samples were then subjected to centrifugation at 5000 rpm for 10 min, and the resulting sera were promptly preserved at −20 °C in a deep freezer until required for subsequent biochemical analyses. Post-blood collection, testicular tissues were carefully extracted. Half of the left testicle tissue was meticulously fixed in a 4% buffered formaldehyde solution using the immersion method, preparing it for histopathological examination. Half of the left testicle tissue was promptly transferred to a −80 °C deep freezer, where it remained until needed for biochemical analyses. This marked the comprehensive conclusion of the research endeavors and the meticulous collection of pertinent samples for further investigation.

### 2.3. Measurement of Antioxidants Enzymes (SOD, GSH) and Lipid Peroxidation (MDA)

SOD enzyme activity, MDA levels, and GSH levels were quantified in testicular tissues. SOD enzyme activity was assessed following the protocol outlined by Sun et al. and reported in units per milligram of protein [15]. This technique relies on the ability of SOD enzyme activity to prevent superoxide anions, the reduced form of the superoxide radical (O_2_^−^), from lowering nitroblue tetrazolium (NBT) dye. MDA content was determined according to the methodology of Placer et al. [16]. The fundamental idea behind this procedure is that MDA and thiobarbituric acid (TBA) combine to generate a colorful complex. GSH content was evaluated using the approach outlined by Sedlak and Lindsay [17]. The results for MDA and GSH are expressed in micromoles per milligram of tissue.

### 2.4. Measurement of Pro-Inflammatory Cytokines

The concentrations of Interleukin-1β (IL-1β), Interleukin-6 (IL-6), and Tumor Necrosis Factor-α (TNF-α) in the testis tissues were measured using a rat ELISA kit (Sunred Biological Technology, Shanghai, China) based on the manufacturer’s guidelines. The absorbance was recorded at 450 nm with an ELISA microplate reader (Bio-Tek, Winooski, VT, USA).

### 2.5. Immunoblotting

Testis tissue samples were weighed, crushed with nitrogen gas, and then homogenized in RIPA buffer with inhibitors (Ecotech Bio, Erzurum, Turkey). To examine the protein expressions of TLR4, caspase-3, Bcl-2, NRF-2, HO-1, PI3K, mTOR, and AKT-1, the homogenate was processed using a tissue lyser (Qiagen, Germantown, MD, USA). Using a Pierce BCA assay (Thermo Sci., St. Bend, IN, USA), the protein concentration was measured. An amount of 30 µg of protein was transferred to a PVDF membrane following SDS-PAGE (10%), blocked with 5% BSA for 90 min, and then incubated with primary antibodies at 4 °C for the entire night. Protein bands were found using ECL (Thermo, West Chester, PA, USA, 3405) and examined using Image LabTM Software for Mac Version 6.1 (Bio-Rad, Hercules, CA, USA) after the membranes had been cleaned and treated with secondary antibodies (Santa Cruz, sc-2004/sc-2005) for 90 min.

### 2.6. Epididymal Spermatozoon Density Measurement

After being dissected, the left epididymis was put in a Petri plate with 1 milliliter of a 0.9% NaCl solution. To guarantee complete disruption, the epididymal tissue was gently crushed for two minutes using pliers. To help the spermatozoa release into the saline, the homogenized mixture was incubated for four hours at room temperature. The spermatozoa suspension was pulled up to the 0.5 mark on a red blood cell pipette following incubation. An amount of 5 g of sodium bicarbonate, 100 mL of distilled water, 1 mL of formalin, and 25 mg of eosin were dissolved to create an eosin solution, which was then added to the pipette until the volume reached 101, resulting in a 1:200 dilution.

With coverslips already attached, roughly 10 µL of this diluted sample was added to each chamber of a Thoma slide (depth: 0.1 mm, area: 0.0025 mm^2^). To guarantee that the spermatozoa were evenly distributed over the counting area, the slide was put under a light microscope and left to stand for five minutes. A light microscope with 200× magnification was used to count the spermatozoa in each square of the two counting zones [18].

### 2.7. Spermatozoon Motility Assessment

The heating table of the microscope was used to position a microscope slide, which was then left to stabilize at 37 °C. The slide was then placed on the heating table and covered with 200 µL of Tris buffer solution, which is made up of 3.63 g Tris (hydroxymethyl) aminomethane, 0.50 g glucose, 1.99 g citric acid, and 100 mL deionized water. A 5–10 µL suspension containing spermatozoa, obtained from the left cauda epididymis via sectioning, was added to the Tris buffer solution. The suspension was homogenized by gentle mixing with a coverslip. Sperm motility was manually assessed under a light microscope at 400× magnification. For motility estimation, semen was directly collected from the cauda epididymis. Three fields of one drop of suspension were examined, and the average motility rate from these fields was calculated to determine the percent motility [19].

### 2.8. Abnormal Spermatozoon Rate Determination

To assess the abnormal spermatozoon ratio, 20 µL of the spermatozoon mixture in tris buffer, which had been used for motility evaluation, was dropped onto a slide preheated to 37 °C. Two drops of eosin–nigrosin dye solution were added to the mixture and homogenized. After the smear was dried, it was examined under a light microscope at 400× magnification. In total, 200 spermatozoa were analyzed per smear, and the percentage of abnormal spermatozoa was calculated based on this examination [19].

### 2.9. Histological Procedures Evaluation of Testicular Tissue

The testes were cleaned with running tap water after fixation, dehydrated using a series of graded alcohols (70%, 80%, 90%, 96%, 100%), and deparaffinized using a series of xylene. They were then incubated in three separate paraffin baths at 60 °C for embedding. After 5-micron-thick sections were taken, the tissue sections were mounted on ground-frozen slides for hematoxylin and eosin staining and poly-l-lysine-coated slides for histopathological preparations. Analysis was performed on the stained sections under a light microscope (Nikon Eclipse, Shanghai, China) for histological evaluation. For each sample, 7 to 8 consecutive tissue sections were analyzed, and 15 to 20 seminiferous tubules were chosen at random for assessment from every section. The testicular tissues were evaluated for histopathological and spermatogenic features using the Johnsen scoring system [20], which involves 10 specific histological parameters. Using these parameters as the Score Criteria, Seminiferous tubule histological structure was assessed using a 10-point rating system. Whereas Score 2 denotes the presence of solely Sertoli cells and no germ cells, Score 1 denotes the absence of cells in the tubular sections. Score 4 shows that there are just a few spermatocytes present, while Score 3 shows solely spermatogonia. A score of five indicates the presence of many or many spermatocytes but the absence of spermatozoa and spermatids. A score of 6 indicates that there are very few spermatids present, whereas a score of 7 indicates that there are many spermatids but no spermatozoa. While Score 9 describes chaotic spermatogenesis despite the presence of many spermatozoa, score 8 indicates the existence of only a few spermatozoa, and a score of 10 denotes the highest degree of normalcy and full spermatogenesis with well-structured tubules. The average Johnsen score was determined for each sample to quantify spermatogenic activity [20]. The analyses were performed by a histologist who was blinded to the identities of the samples, utilizing a coding system to ensure objective and unbiased results. Histopathological damage was assessed using four different histological criteria form the basis of the Cosentino classification [21] (Table 1). To guarantee a thorough evaluation, 50 seminiferous tubules were examined for scoring purposes per section.

### 2.10. Statistical Analyses

Normality tests were performed to determine the appropriate statistical analysis method (parametric or non-parametric) for the study. Specifically, the Kolmogorov–Smirnov and Shapiro–Wilk tests were utilized. Given that the *p*-values were less than 0.05, it was determined that the data deviated from a normal distribution finding showed that non-parametric tests were chosen as substitutes for parametric methods. We used the Mann–Whitney U test for biochemical parameters. Immunostaining H-scores were compared using Tukey’s HSD, a post hoc test for one-way ANOVA, after the homogeneity of variances test. Statistical significance is defined as a *p*-value of less than 0.05, and all data are presented as mean ± standard deviation (mean ± SD). We used SPSS 22.0 to analyze the data.

## 3. Results

### 3.1. Effect of Bro on Antioxidant Enzymes and Lipid Peroxidation in I/R-Induced Testis Injury

In Figure 1, the effects of Bro treatments on enzymatic activities of SOD, MDA levels, and GSH levels in testicular tissue are displayed. The findings demonstrate that ischemia–reperfusion (I+R) markedly increases MDA, a marker of lipid peroxidation, and significantly reduces antioxidant defense systems, including reduced GSH and SOD activity. The group treated with Bro alone showed an increase in MDA levels, while the increase in the ischemia and I+R groups was significantly greater (Figure 1a). In contrast to the I+R group, Bro administration in the I+R+Bro group reduced the MDA level (Figure 1a) and restored GSH (Figure 1b) and SOD (Figure 1c) activity. These findings suggest that Bro may have a therapeutic role in lowering I/R injury since it reduces oxidative stress and enhances antioxidant status.

### 3.2. Effect of Bro on Pro-Inflammatory Cytokines Production in I/R-Induced Testis Injury

Figure 2 illustrates the protective effect of Bro and the inflammatory response associated with I/R injury in the testis. The I+R group exhibited a notable increase in inflammatory indicators, like TNF-α (Figure 2b), IL-1β (Figure 2a), and IL-6 (Figure 2), compared to the control group. These high levels indicate a severe inflammatory reaction brought on by oxidative stress and I+R-induced tissue damage. The anti-inflammatory impact of bro therapy was demonstrated by the significantly lower TNF-α (Figure 2b), IL-1β (Figure 2a), and IL-6 (Figure 2c) levels in the I+R+Bro group as compared with those in the I+R group. Bro’s antioxidant qualities and ability to inhibit cytokine synthesis may be responsible for its ability to lessen the inflammatory cascade. These inflammatory markers did not rise in the Bro-alone group, indicating its safety. According to these results, Bro may be used as a treatment for diseases marked by ischemia–reperfusion injury and related inflammatory reactions since it not only lowers oxidative stress but also regulates inflammation.

### 3.3. Effect of Bro on Apoptosis and Inflammatory Biomarkers in I/R-Induced Testis Injury

Our results demonstrate the role of Bro in modulating apoptosis and inflammation during I/R injury in the testis. Western blot analysis (Figure 3a) and relative protein density measurements (Figure 3b–d) reveal significant changes in key proteins associated with cell survival, apoptosis, and inflammation. Bcl-2, an anti-apoptotic protein, significantly decreased in the groups with ischemia and I+R, reflecting increased susceptibility to apoptosis.

However, Bro co-treatment (I+R+Bro) partially restored Bcl-2 levels, suggesting its protective role in enhancing cell survival by inhibiting apoptosis (Figure 3b). Additionally, a crucial apoptotic executor, caspase-3, showed elevated expression in the ischemia and I+R groups, indicating enhanced apoptotic activity. Treatment with Bro significantly reduced caspase-3 levels in the I+R+Bro group, confirming its anti-apoptotic effect (Figure 3c). Moreover, toll-like receptor 4 (TLR4), an upstream regulator of inflammation and apoptosis, was upregulated in the ischemia and I+R groups, reflecting heightened inflammatory signaling. Indeed, Bro treatment significantly suppressed TLR4 expression in the I+R+Bro group, demonstrating its anti-inflammatory properties. These findings suggest that Bro treatment modulates apoptosis and inflammation by upregulating Bcl-2 while suppressing caspase-3 and TLR4. This dual effect highlights Bro’s therapeutic potential in reducing cellular injury and inflammatory damage in ischemia–reperfusion conditions.

### 3.4. Bro Treatment Restores Nrf2 and HO-1 Levels in I/R-Induced Testis Injury

Figure 4 demonstrates HO-1 and Nrf2’s protein expression under various experimental conditions, including control, Bro, Ischemia, I+R, and I+R+Bro. Figure 4a displays HO-1, Nrf2, and β-actin (loading control) sample Western blot bands. Figure 4b,c quantify the relative protein density of HO-1 and Nrf2, respectively, normalized to β-actin. Under ischemic and I+R conditions, both HO-1 and Nrf2 protein levels were significantly reduced, indicating oxidative damage and suppressed antioxidant defense pathways. Indeed, treatment with Bro (I+R+Bro) significantly restored HO-1 and Nrf2 expression compared to I+R alone. On the other hand, the Bro-alone treatment showed similar expression levels to the control, suggesting no adverse effects in basal conditions but potential antioxidant properties. Taken together, these data suggest Bro’s protective effect against the oxidative stress brought on by I/R damage, possibly through the Nrf2-mediated upregulation of HO-1, a critical antioxidant enzyme.

### 3.5. Bro Treatment Promotes Cell Survival Under I/R-Induced Testis Injury

AKT, mTOR, and PI3K protein expression are shown in Figure 5 under several circumstances, including control, Bro-only therapy, ischemia, I+R, and I+R+Bro. Western blot bands for AKT, mTOR, PI3K, and β-actin (loading control) are depicted in Figure 5a. Figure 5b–d shows the relative protein densities for PI3K, mTOR, and AKT. During ischemia, mTOR and PI3K expression levels were much higher than control, which most likely suggests that the PI3K/AKT/mTOR signaling pathway was compensated for by stress. However, AKT levels were decreased compared to mTOR and PI3K, indicating that ischemia might have interfered with the upstream signaling of the pathway. The I+R condition further enhanced mTOR, PI3K, and AKT expression significantly increased compared to control or Bro, demonstrating the PI3K/AKT/mTOR pathway’s hyperactivation, which is frequently connected to maladaptive stress reactions.

Remarkably, this route was regulated by the addition of Bro (I+R+Bro). When compared to I+R alone, levels of AKT in the I+R+Bro group were considerably normalized, indicating that Bro had a regulatory function. Comparing the I+R+Bro group to I+R alone, PI3K and mTOR levels were also marginally lower, suggesting that Bro extract lessens excessive pathway activation. According to these findings, Bro therapy promotes the PI3K/AKT/mTOR pathway’s regulated activation, which helps cells recover and maintain homeostasis during ischemia–reperfusion.

### 3.6. Effect of Bro on Spermatologic Examinations in I/R-Induced Testis Injury

Table 2 displays the spermatologic information for the weighted reproductive organs (ventral prostate, seminal glands, epididymides, and testes). Regarding sperm motility and density, there was no statistically significant difference between the control group and the Bro group. (*p* < 0.05). The I/R group’s sperm motility, density, and aberrant sperm ratio differed considerably from both the control and Bro groups (*p* ˂ 0.001). In contrast with the I/R group, the sperm parameters of the I/R + Bro group were significantly better (*p* < 0.001).

### 3.7. Histologic Evaluation of Testicular Tissue

Upon examining histologic sections of the sham group, many seminiferous tubules with uniform outlines and normal interstitial space were found. Examining the seminiferous tubules revealed that Sertoli cells have a pyramidal shape with pale nuclei sitting on the basement membrane, as well as cells at various phases involved in spermatogenesis. The lumen of the seminiferous tubule was seen to be filled with the flagella of mature spermatids. Examining the intertubular regions revealed Leydig cells. Round spermatids, elongated spermatids, main and secondary spermatocytes, and spermatogonia produced by active spermatogenesis comprised the germinative epithelium (Figure 6). Necrotic seminiferous tubules and interstitial bleeding were seen in the histological sections of the ischemia and I+R groups (Figure 6).

The seminiferous tubules’ morphology was uneven, and the basement membrane was degenerating. In the seminiferous tubules, the germinative epithelium was seen to have separated from the basement membrane and exfoliated into the lumen (Figure 6). Furthermore, congestion in the arteries and edema in the interstitial space were noted. Examining the histological sections of the I+R+Bro group revealed that there were local interstitial hemorrhages, the interstitial space was largely normal, and the seminiferous tubules were more regular than in the I+R group.

Examining the seminiferous tubules revealed that the germinative epithelium was made up of cells with varying stages of spermatogenesis. The basement membrane was more regular in the control (sham) group than in the I+R group, and the Sertoli cells’ nuclei rested on it. The seminiferous tubules’ epithelium showed signs of degenerative alterations and epithelial shedding. The intertubular regions were shown to contain Leydig cells. The seminiferous tubules exhibited germinative cells exhibiting active spermatogenesis, much like the control (sham) group, and the lumen of the tubules was packed with mature spermatid flagella.

There was no discernible differentiation between the group’s ischemia and I/R or between the sham group and the Bro group in statistical comparisons across groups using the modified Johnsen rating criteria (Figure 7a). Nonetheless, the difference between the groups I+R and I+R+Bro was statistically significant (Figure 7a). The I+R+Bro and sham groups did not have a difference that is statistically significant. The sham and Bro groups showed no histological results in comparisons based on the Cosentino categorization for evaluating histopathological damage. The differences between the I and I+R groups were not statistically significant, as seen in Figure 7b. The I+R and I+R+Bro groups did, however, differ statistically significantly (Figure 7b).

## 4. Discussion

Severe oxidative stress is a hallmark of I/R injury, which causes substantial cellular damage in a variety of tissues. When blood flow is restored to ischemic areas, this event takes place, leading to elevated ROS levels, which are linked to the pathophysiology of testicular injury [22,23]. Important enzymes like xanthine oxidase play a vital part in the production of ROS during I/R events, which is connected to elevated MDA levels, a symptom of lipid peroxidation [23,24]. The presence of oxidative damage in tissues as a marker of lipid peroxidation is confirmed by our study’s considerably elevated MDA levels in the I and I+R groups. By inhibiting lipid peroxidation, Bro may protect testicular tissue, as seen by the considerable decrease in this rise in MDA concentrations in the I+R+BRO group that received Bro pretreatment. These conclusions are supported by research on Bro’s antioxidant qualities, particularly its capacity to prevent lipid peroxidation [25,26]. An increase in MDA levels observed in the Bro-treated group can be evaluated as an indicator of oxidative stress. However, the absence of a significant change in antioxidant parameters such as GSH and SOD indicates that cellular defense mechanisms are sufficiently activated or a compensatory response occurs. Studies have reported that some pharmacological agents may affect specific biochemical parameters differently [27,28]. Therefore, the effects of Bro on oxidative balance have been limited to the impact on the cellular defense system. A metalloenzyme called SOD scavenges superoxide radicals and is essential for preventing oxidative stress. SOD activity declines during I/R injury, and GSH, another essential endogenous antioxidant, may also be reduced [29].

According to published reports, testicular torsion results in significant histopathologic alterations in the ipsilateral testicle and structural and functional degeneration in the contralateral testicle [30]. Histopathologic features such as tubular degeneration, loss of germ cells, and interstitial edema are common among these alterations [1]. According to the literature, the mechanism of the contralateral testis being damaged may be more closely linked to immunological reactions, elevated oxidative stress, and systemic inflammatory reactions, depending on the degree and kind of damage between the two testes [31]. ROS and pro-inflammatory cytokines, produced in the bloodstream due to reperfusion injury in the ipsilateral testis, might specifically impact the contralateral testis through the bloodstream and result in histopathologic alterations [32]. Moreover, cellular injury in the contralateral testis may potentially be caused by autoimmune processes [31].

In the I and I/R groups of our study, GSH and SOD levels dramatically dropped, indicating compromised antioxidant defenses due to oxidative stress brought on by I/R. Our results are consistent with past research showing that testicular tissue is particularly vulnerable to harm brought on by free radicals. [33,34]. The rise in these levels in the I/R+Bro group, however, indicates that Bro may help preserve enzyme activity and shield testicular tissue from I/R damage. An elevated inflammatory response is frequently linked to ischemia/reperfusion (I/R) damage [35]. The I and I+R groups in this research had markedly elevated levels of TNF-α, IL-6, and IL-1β, implying inflammatory processes were triggered. It is commonly known that these cytokines cause oxidative stress and cellular damage brought on by I/R injury [36]. Specifically, the elevated levels of IL-1β in the I+R group suggest that the initial inflammatory response is intensified, leading to significant harm to the testicular tissue. Nonetheless, the I+R+Bro group’s levels of TNF-α, IL-6, and IL-1β were much lower, implying that Bro has an anti-inflammatory effect. This implies that Bro’s anti-inflammatory qualities modulate the I/R-induced inflammatory response, hence reducing testicular tissue damage.

As reported in the literature [37], Bro may suppress pro-inflammatory cytokines through intracellular signaling pathways by inhibiting the production of inflammatory mediators. In the study, we detailed the effect of Bro against I/R injury on the expression levels of cellular response proteins like Bcl-2, caspase-3, TLR4, HO-1, NRF2, AKT1, mTOR, and PI3K. Firstly, a reduction in the expression of Bcl-2 in the I and I+R groups indicates that apoptotic processes are activated with a decrease in this protein that inhibits cell death. In contrast, increased levels of caspase-3 and TLR4 confirm the enhancement of cell death and inflammatory response. These findings suggest that I/R causes severe damage to testicular tissue by triggering apoptosis in cells. However, increased expression of Bcl-2 and reduced levels of Caspase-3 and TLR4 expression in the I+R+Bro group suggest that Bro protects cell health by suppressing apoptotic and inflammatory processes. These results support the tissue protection potential of Bro [38] through the regulation of apoptosis.

The I and I+R groups’ lower levels of antioxidant defense systems HO-1 and NRF2 suggest that oxidative stress cannot be better controlled [39]. These proteins are crucial in the defense against oxidative stress, particularly the NRF2 pathway [40]. The increased expression of these proteins in the I+R+Bro group suggests that Bro alleviates oxidative stress and contributes to tissue protection potential by activating antioxidant defense mechanisms. AKT1, mTOR, and PI3K signaling pathways, which play central roles in cellular growth, survival, and metabolic regulation processes [41], were found to be increased in I and I+R groups. I/R injury may lead to the activation of these pathways by triggering cellular stress responses [42], which may be considered a reflection of the tissue’s attempts to adapt to stress. Excessive PI3K/AKT/mTOR signaling pathway activation can cause long-term cellular dysfunction and disturb cellular homeostasis, even though it normally promotes cell growth, proliferation, and survival [43]. Given this, the elevated levels of AKT1, mTOR, and PI3K expression in the I and I+R groups imply that, in response to stress, cells over-activate this signaling pathway, inciting pathophysiological events. The stabilizing impact of Bro on cellular stress is demonstrated by the inhibition of these signaling pathways in the I+R+Bro group. By inhibiting excessive cell growth and proliferative responses, inhibition of the PI3K/AKT/mTOR pathway preserves energy balance and may lessen damage brought on by oxidative stress.

Over-activation of this system has been linked in the literature to apoptosis and oxidative stress. One significant defense mechanism against I/R-induced damage may be the suppression of these pathways, which protects Bro cells. There was no statistical difference between the control group and the Bro group in terms of sperm motility and density (*p* < 0.05). However, the I+R group’s sperm motility, density, and abnormal sperm ratio differed significantly from those of the control and Bro groups, indicating that I+R injury impairs spermatogenesis and lowers sperm quality. This finding supports the body of research on the detrimental effects of oxidative stress on reproductive health [13,44]. The I+R+Bro group’s sperm parameters significantly improved in contrast to those of the I+R group, which implies that Bro might have protective benefits against testicular damage brought on by I/R. Bro’s potential effects could be explained by its mechanisms that promote cellular endurance or its antioxidant activity. Accordingly, the increase in sperm motility and density in the I+R+Bro group raises the possibility that Bro could be a useful treatment for lessening testicular damage brought on by ischemic/reperfusion episodes.

Necrotic seminiferous tubules and interstitial hemorrhage, seminiferous tubule distortion, and basement membrane degradation were noted in the histological sections of the ischemia and I/R groups. The germinative epithelium was seen to have shed into the seminiferous tubule lumen after separating from the basement membrane. Furthermore, congestion of the arteries and edema in the interstitial space was found. These findings are consistent with those found in the body of current literature and demonstrate the harm that I/R injury causes to the testicles [45]. Seminiferous tubules were more regular, the interstitial space was normal, and there were occasional hemorrhages in the I+R+Bro group. The basement membrane appeared more regular, and the germinative epithelium contained cells at spermatogenesis stages closer to the control (sham) group. The I+R and I+R+Bro groups differed significantly from one another based on the modified Johnsen scoring criteria; however, between the Control (sham) and Bro groups, there were no notable differences. The histological results support Bro’s ability to repair testicular damage brought on by I/R.

## 5. Conclusions

This research showed the protective effects of Bro in mitigating I/R injury in testicular tissue, highlighting its potential as a therapeutic agent. Bro treatment significantly reduced oxidative stress and inflammation associated with I/R-induced damage, while also enhancing the activity of pivotal cellular response proteins linked to apoptosis and antioxidant defense mechanisms. The improvements in sperm parameters and the histological integrity of the seminiferous tubules strongly support Bro’s ability to mitigate the detrimental effects of I/R injury. Overall, the protective effects of Bro can be attributed to its antioxidant properties in combating oxidative stress and its capacity to modulate cellular stress responses, including the suppression of signaling pathways associated with excessive cellular growth and apoptosis. These findings underscore the need for further investigation into the protective mechanisms of Bro and its potential clinical applications. The use of a single dose, possible species-specific variations, and the lack of long-term outcome evaluations are some of the study’s shortcomings that should be taken into account when interpreting the results. Future studies should explore a broader dose range to optimize Bromelain’s therapeutic potential in testicular I/R injury and employ specific inhibitors or knockout models to validate its mechanistic pathways. Additionally, complementary in vitro research using pathway-specific inhibitors and cell models is needed to further elucidate the underlying molecular mechanisms.

## Figures and Tables

**Figure 1 biomolecules-15-00292-f001:**
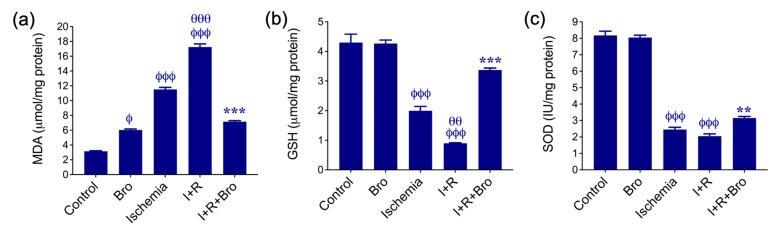
Antioxidant parameters of testis tissues across experimental groups. MDA levels (**a**), GSH levels (**b**), and SOD activity (**c**) in testis tissue subjected to I/R injury. The mean ± SD represents the findings from three different experiments. Statistical significance is represented as follows: ^ϕ^ *p* < 0.05 and ^ϕϕϕ^ *p* < 0.01 for comparisons between the control and Bro, ischemia, and I+R groups; ^θθ^ *p* < 0.01 and ^θθθ^ *p* < 0.001 for the ischemia vs. I+R group; and ** *p* < 0.01 and *** *p* < 0.001 for the I+R vs. I+R+Bro group. Bars without a phi, theta or asterisk symbol (*) denote a lack of statistical significance.

**Figure 2 biomolecules-15-00292-f002:**
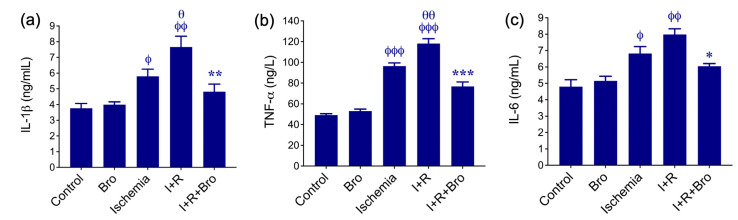
Pro-inflammatory cytokine levels in serum across experimental groups: IL-1β (**a**), TNF-α (**b**), and IL-6 (**c**) levels in testis tissue subjected to I/R injury. The mean ± SD represents the findings from three different experiments. Significance in statistics: ^ϕ^ *p* < 0.05, ^ϕϕ^ *p* < 0.01, and ^ϕϕϕ^ *p* < 0.001 of the control vs. Bro, ischemia, and I+R groups; ^θ^ *p* < 0.05 and ^θθ^ *p* < 0.01 of the ischemia vs. I+R group; and * *p* < 0.05, ** *p* < 0.01, and *** *p* < 0.001 of the I+R vs. I+R+Bro group. Bars that do not display a phi (ϕ), theta (θ), or asterisk (*) symbol represent no statistical significance.

**Figure 3 biomolecules-15-00292-f003:**
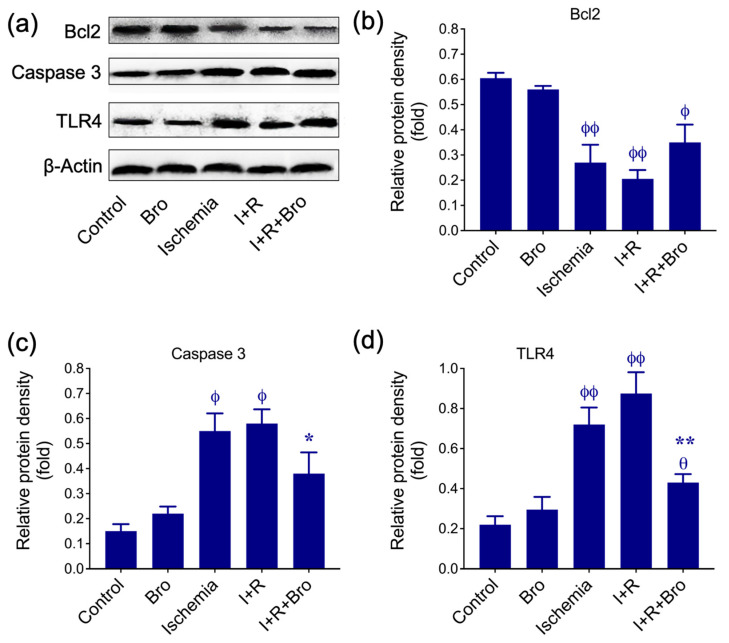
(**a**) Testicular tissues’ levels of Bcl-2, caspase-3, and TLR4 protein expression are shown. Results are normalized to the internal reference GAPDH. (**b**–**d**) The histograms demonstrate the relative protein density of various target proteins. Three separate experiments’ worth of data are presented as mean ± SD. Significance in statistics: * *p* < 0.05, ** *p* < 0.01 for the I+R vs. I+R+Bro group; ^θ^ *p* < 0.05 a for the ischemia vs. I+R and I+R+Bro groups; and ^ϕ^ *p* < 0.05 and ^ϕϕ^ *p* < 0.01 for the control vs. Bro, ischemia, I+R, and I+R+Bro groups. The absence of statistical significance is shown by bars that do not have an asterisk (*) or theta (θ) or phi (ϕ) symbol. The original images of the Western bolt can be found in the Appendix A.

**Figure 4 biomolecules-15-00292-f004:**
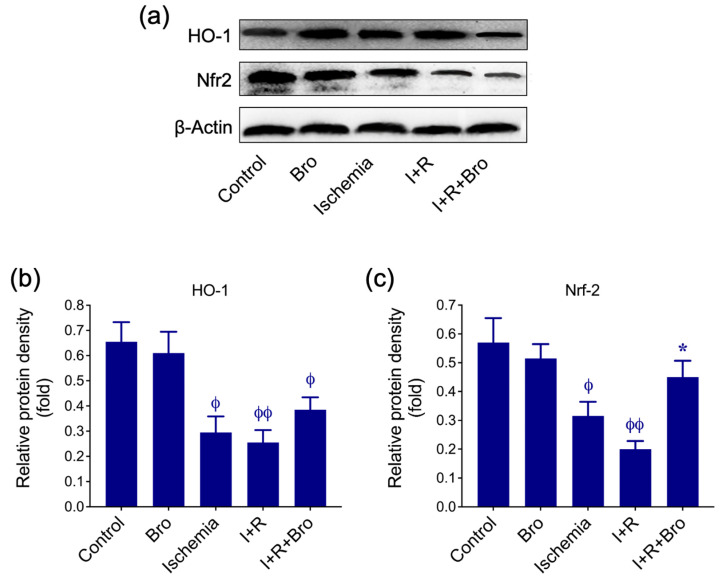
(**a**) HO-1 and Nrf2 ’s protein expression levels in testicular tissues are shown. (**b**,**c**) Histograms represent the relative protein density of the target proteins, standardized to the GAPDH internal control. The mean ± SD of three separate experiments is used to express the data. Significance in statistics: ^ϕ^ *p* < 0.05 and ^ϕϕ^ *p* < 0.01 of the control vs. Bro, ischemia, I+R, and I+R+Bro groups; * *p* < 0.05 of the I+R vs. I+R+Bro group. Bars that do not display a phi (ϕ) or asterisk (*) symbol represent no statistical significance.

**Figure 5 biomolecules-15-00292-f005:**
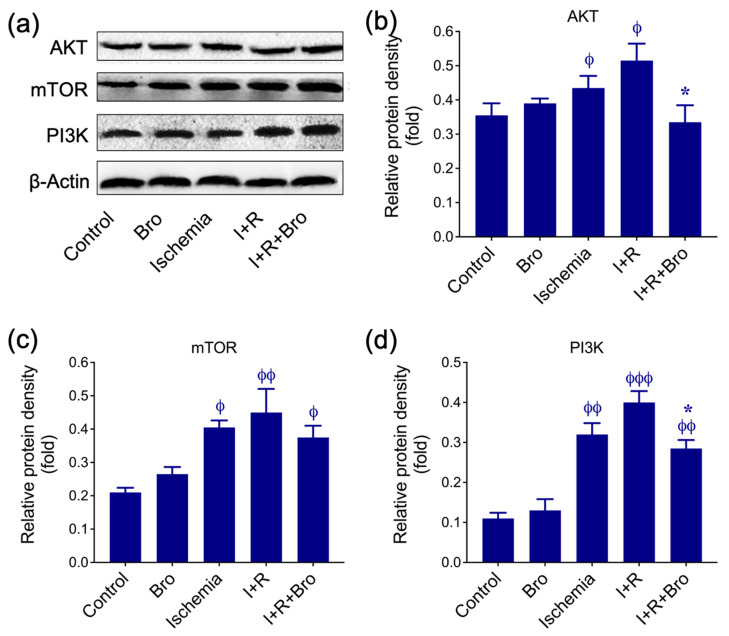
(**a**) Protein expression levels for AKT, mTOR, and PI3K in testicular tissues. (**b**–**d**) Histograms depict the relative protein density of target proteins, normalized to the internal control GAPDH. Data from three distinct experiments are displayed as mean ± SD. Statistical significance: ^ϕ^ *p* < 0.05, ^ϕϕ^ *p* < 0.01, and ^ϕϕϕ^ *p* < 0.001 of the control vs. Bro, ischemia, I+R, and I+R+Bro groups; * *p* < 0.05, of the I+R vs. I+R+Bro group. Bars lacking a phi (ϕ) or asterisk (*) symbol indicate that there is no statistical significance.

**Figure 6 biomolecules-15-00292-f006:**
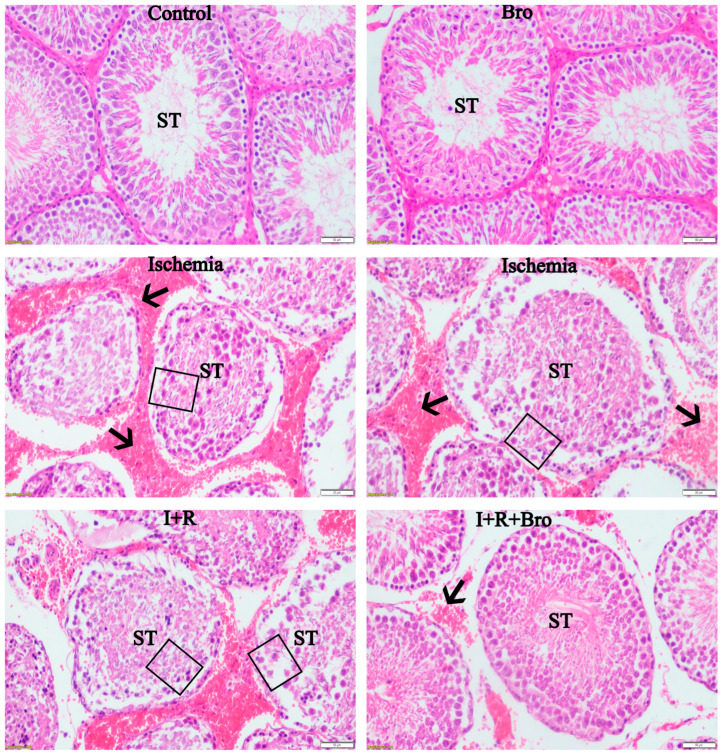
Testicular tissue histological visualization by experimental group: The ischemia, I+R, and I+R+Bro groups exhibit notable changes in histological appearance, while the control (sham) and Bro groups have normal histological appearances. ST represent seminiferous tubules, squares represent necrotic seminiferous tubules and seminiferous tubule cell exfoliation, whereas arrows represent interstitial hemorrhage. Staining with hematoxylin and eosin, 200× magnification.

**Figure 7 biomolecules-15-00292-f007:**
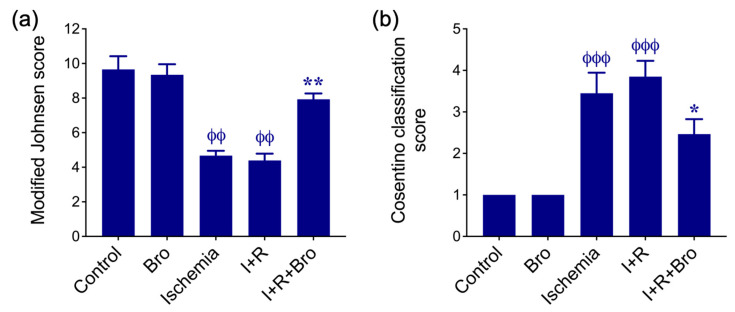
(**a**) Modified Johnsen scores of all the groups. (**b**) Cosentino classification scores of all the groups. The data from three separate experiments are presented as mean ± SD. Statistical significance: ^ϕϕ^ *p* < 0.01, and ^ϕϕϕ^ *p* < 0.001 of the control vs. Bro, ischemia, I+R, and I+R+Bro groups; * *p* < 0.05 and ** *p* < 0.01 of the I+R vs. I+R+Bro group. Bars that lack a phi (ϕ) or asterisk (*) symbol signify no statistical significance.

**Table 1 biomolecules-15-00292-t001:** Cosentino classification of histopathological damage.

Grade	Criteria
Grade 1	Normal germinal cells in the testicles
Grade 2	Mild hemorrhage interstitial edema, or less regular, densely packed seminiferous tubules
Grade 3	Widespread hemorrhage an irregular pyknotic nucleus, and less pronounced seminiferous tubule borders
Grade 4	Tightly packed seminiferous tubules with germ cell coagulation necrosis

**Table 2 biomolecules-15-00292-t002:** Semen parameters findings of all groups.

Parameters	Control	Bro	Ischemia	I+R	I+R+Bro
Total Motility (%)	72.22 ± 11.08	76.12 ± 3.33	44.66 ± 28.73 ^ϕ^	12.85 ± 17.47 ^ϕ,θ^	60.00 ± 28.08 ^ϕ,^*
Concentration (×10^6^/cauda)	82.25 ± 7.47	84.33 ± 3.50	63.50 ± 6.51 ^ϕ^	47.92 ± 7.21 ^ϕ,θ^	83.33 ± 4.16 ^ϕ,^*
Abnormal Head Spermatozoon	4.20 ± 0.83	4.16 ± 1.32	5.00 ± 2.82	5.42 ± 2.07	5.66 ± 2.08
Abnormal Tail Spermatozoon	9.20 ± 3.34	10.16 ± 4.11	10.21 ± 3.27	14.28 ± 3.77	8.66 ± 2.08
Abnormal Total Spermatozoon	13.40 ± 3.91	14.33 ± 5.20	15.21 ± 5.89	19.71 ± 5.64	14.33 ± 4.16
Right Testis (gr)	1.48 ± 0.14	1.43 ± 0.09	1.36 ± 0.05	1.49 ± 0.17	1.42 ± 0.05
Left Testis (gr)	1.68 ± 0.12	1.71 ± 0.16	1.37 ± 0.40 ^ϕ^	1.45 ± 0.10 ^ϕ,θ^	1.54 ± 0.20 ^ϕ,^*
Right Epidydimis (gr)	0.62 ± 0.11	0.60 ± 0.08	0.45 ± 0.13	0.50 ± 0.06	0.45 ± 0.05
Left Epidydimis (gr)	0.61 ± 0.06	0.62 ± 0.09	0.47 ± 0.22 ^ϕ^	0.48 ± 0.07 ^ϕ^	0.50 ± 0.11 *
Right Cauda (gr)	0.24 ± 0.07	0.27 ± 0.03	0.17 ± 0.03	0.24 ± 0.08	0.24 ± 0.10
Vesicula Seminalis (gr)	0.83 ± 0.12	0.93 ± 0.20	0.65 ± 0.11 ^ϕ^	0.62 ± 0.11 ^ϕ,θ^	0.63 ± 0.09
Prostate (gr)	0.22 ± 0.02	0.22 ± 0.18	0.28 ± 0.06 ^ϕ^	0.32 ± 0.18 ^ϕ,θ^	0.28 ± 0.00

Three separate experiments’ mean ± SD show the data are presented. Statistical significance: * *p* < 0.05 for the I+R vs. I+R+Bro group; ^θ^ *p* < 0.05 a for the ischemia vs. I+R and I+R+Bro groups; and ^ϕ^ *p* < 0.05 for the control vs. Bro, ischemia, I+R, and I+R+Bro groups. There is no statistical significance shown by bars that lack the phi, theta, or asterisk symbols.

## Data Availability

Data available on request from the authors.

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
