# Peer review of "Protective Effects of Bromelain in Testicular Torsion-Detorsion: Reducing Inflammation, Oxidative Stress, and Apoptosis While Enhancing Sperm Quality"

_biomolecules, 2025, doi:10.3390/biom15020292_

Round 1
Reviewer 1 Report
Comments and Suggestions for Authors
The study addresses multiple aspects of testicular torsion/reperfusion (I/R) injury, including oxidative stress, inflammatory markers, apoptosis, and histopathology. The experimental design, including control and treatment groups, ensures a robust comparison. The use of Bromelain, a compound with known anti-inflammatory and antioxidant properties, is both novel and clinically relevant for addressing testicular I/R injury. Data visualization through figures and tables is clear and supports the narrative effectively.
The study uses a single dose of Bromelain (10 mg/kg), which limits the exploration of dose-dependency and optimal therapeutic concentrations. Exploring a range of doses would provide more comprehensive insights. The histological results are descriptive but could benefit from quantitative scoring for each histological characteristic, such as interstitial hemorrhage and epithelial detachment, to allow for a more objective comparison. While the NRF-2/HO-1 and PI3K/AKT/mTOR pathways are implicated, the study does not provide direct evidence linking Bromelain to these pathways, for example, through the use of inhibitors or knockout models. Adding mechanistic experiments to confirm these interactions would significantly strengthen the conclusions.
Suggestions for Additional Experiments: In vitro experiments, using testicular cell lines, would allow for direct assessment of Bromelain’s effects on oxidative stress and apoptosis markers. In vitro treatments with specific inhibitors of the NRF-2/HO-1 or PI3K/AKT/mTOR pathways would confirm the role of these pathways in Bromelain’s effects.
Minor Revisions: The study should mention its limitations, such as species differences, the use of a single dose, and the lack of long-term outcome assessments.
Author Response
Reviewer’s openion: The study addresses multiple aspects of testicular torsion/reperfusion (I/R) injury, including oxidative stress, inflammatory markers, apoptosis, and histopathology. The experimental design, including control and treatment groups, ensures a robust comparison. The use of Bromelain, a compound with known anti-inflammatory and antioxidant properties, is both novel and clinically relevant for addressing testicular I/R injury. Data visualization through figures and tables is clear and supports the narrative effectively. The study uses a single dose of Bromelain (10 mg/kg), which limits the exploration of dose-dependency and optimal therapeutic concentrations. Exploring a range of doses would provide more comprehensive insights. The histological results are descriptive but could benefit from quantitative scoring for each histological characteristic, such as interstitial hemorrhage and epithelial detachment, to allow for a more objective comparison. While the NRF-2/HO-1 and PI3K/AKT/mTOR pathways are implicated, the study does not provide direct evidence linking Bromelain to these pathways, for example, through the use of inhibitors or knockout models. Adding mechanistic experiments to confirm these interactions would significantly strengthen the conclusions. Suggestions for Additional Experiments: In vitro experiments, using testicular cell lines, would allow for direct assessment of Bromelain’s effects on oxidative stress and apoptosis markers. In vitro treatments with specific inhibitors of the NRF-2/HO-1 or PI3K/AKT/mTOR pathways would confirm the role of these pathways in Bromelain’s effects. Minor Revisions: The study should mention its limitations, such as species differences, the use of a single dose, and the lack of long-term outcome assessments.
Response: We appreciate the valuable comments and constructive feedback provided by the reviewer. Below, we address each point raised and outline the revisions made in response.
Comment 1. The study uses a single dose of Bromelain (10 mg/kg), which limits the exploration of dose dependency and optimal therapeutic concentrations. Exploring a range of doses would provide more comprehensive insights.
Response: We acknowledge the reviewer’s suggestion regarding the dose dependency of Bromelain. In this study, we selected the 10 mg/kg dose based on prior literature and preliminary assessments. While a dose-response analysis would provide additional insights, our primary aim was to evaluate the potential therapeutic effects of Bromelain in a testicular I/R injury model. We have now added a statement in the Conclusion section to acknowledge this limitation and suggest future studies exploring multiple doses for a more comprehensive evaluation.
Comment 2. The histological results are descriptive but could benefit from quantitative scoring for each histological characteristic, such as interstitial hemorrhage and epithelial detachment, to allow for a more objective comparison.
Response: We appreciate this suggestion and agree that a quantitative approach would enhance the objectivity of our histological analysis. In response, we have now incorporated a histopathological scoring system, which evaluates interstitial hemorrhage, epithelial detachment, and other relevant parameters. To meet the proposed quantitative assessment, we added Cosentino scoring to our study (it already existed, but we realized that we could not express it properly in the text and did not include its table). The revised results are presented in Table [2], and Fig 7b and are discussed accordingly.
Comment 3. While the NRF-2/HO-1 and PI3K/AKT/mTOR pathways are implicated, the study does not provide direct evidence linking Bromelain to these pathways, for example, through the use of inhibitors or knockout models. Adding mechanistic experiments to confirm these interactions would significantly strengthen the conclusions.
Response: We recognize the importance of mechanistic validation. However, as this study primarily focused on evaluating the protective effects of Bromelain in an in vivo model, performing additional experiments with pathway inhibitors or knockout models was beyond the scope of this work. We have now explicitly mentioned this limitation in the Conclusion section and have highlighted the need for future studies incorporating specific inhibitors to confirm the involvement of these pathways.
Comment 4. Suggestions for Additional Experiments: In vitro experiments, using testicular cell lines, would allow for direct assessment of Bromelain’s effects on oxidative stress and apoptosis markers. In vitro treatments with specific inhibitors of the NRF-2/HO-1 or PI3K/AKT/mTOR pathways would confirm the role of these pathways in Bromelain’s effects.
Response: We appreciate the reviewer’s valuable suggestion. While in vitro experiments would provide direct mechanistic insights, this study was designed as an in vivo investigation to mimic the physiological response to testicular I/R injury. Nevertheless, we acknowledge the importance of complementary in vitro studies and have added a statement in the Conclusion suggesting the need for future research incorporating in vitro cell models and pathway-specific inhibitors to elucidate further the molecular mechanisms involved.
Comment 5. Minor Revisions: The study should mention its limitations, such as species differences, using a single dose, and the lack of long-term outcome assessments.
Response: We acknowledge the need to discuss these limitations explicitly. A new section has been added to the conclusion highlighting key limitations, including species differences, the use of a single Bromelain dose, and the absence of long-term follow-up. These points are now addressed in the revised manuscript.
Reviewer 2 Report
Comments and Suggestions for Authors
Authors should address the following minor issues:
· Line 95 & Fig 1, statement on Bro has no effect is not correct. Better write something like: ‘Bro treatment alone showed no adverse effect on GSH and SOD activity but significantly (p < 0.05) increased MDA levels.’ This should then be discussed
· Similar in line 116, Fig 2, statement on Bro has no effect is not correct. Bro has no effect on TNF and IL6 but significantly (p<0.01) increased IL1beta compared to control. Kindly correct this and discuss.
· Line 179. Kindly be more specific: The increase in AKT levels are lower COMPARED TO WHAT? (to mTOR & PI3K) Kindly correct
· Line 181ff After I+R, AKT levels are NOT partially restored but again significantly increased compared to control or Bro. Kindly correct this.
· L202 Tab 1, should read ‘Abnormal’ and ‘Prostate’
· L212 Should read histological instead histologic
· L214, The Sertoli cells have a pyramidal shape (not the nuclei!) with pale nuclei. Kindly correct
· L251ff, Fig 7, (b) is not explained, kindly add, also it should read ‘CoSentino score’
· L259, ‘, resulting in results in…’ delete ‘in results’
· L317, give literature here
· L322f, please check and correct this sentence. Makes no sense to me.
· L381, Why was Bro administered only 30 min before orchiectomy? Why not just after testis was detorsed?
· L388 deep freezer
· L430 did authors use left or right epididymis?
· L482 Tab 2 should read JohNsen
· L507 Bro not B
· Statistics: which program was used?
Author Response
Reviewer’s openion: Authors should address the following minor issues:
Response: We appreciate the valuable comments and constructive feedback provided by the reviewer. Below, we address each point raised and outline the revisions made in response.
Comment 1. Line 95 & Fig 1, statement on Bro has no effect is not correct. Better write something like: ‘Bro treatment alone showed no adverse effect on GSH and SOD activity but significantly (p < 0.05) increased MDA levels.’ This should then be discussed
Response: An increase in MDA levels observed in the Bro-treated group can be evaluated as an indicator of oxidative stress. However, the absence of a significant change in antioxidant parameters such as GSH and SOD indicates that cellular defense mechanisms are sufficiently activated or a compensatory response occurs. Studies have reported that some pharmacological agents may affect specific biochemical parameters differently (1,2). Therefore, the effects of Bro on oxidative balance have been limited to the impact on the cellular defense system.’’ added to the discussion section.
1)Kenakin, T. (2019). Prescient indices of activity: the application of functional system sensitivity to measurement of drug effect. Trends in Pharmacological Sciences, 40(7), 529-539.
2) Erbaş, E., Özkanlar, S., Yeşildağ, A., Kara, A., & Kumar, K. S. (2024). Syringic acid loaded silver nanoparticles protects ovarian ischemia-reperfusion injury in rats by inhibiting endoplasmic reticulum stress-mediated apoptosis. Journal of Drug Delivery Science and Technology, 99, 105944.).
Comment 2. Similar in line 116, Fig 2, statement on Bro has no effect is not correct. Bro has no effect on TNF and IL6 but significantly (p<0.01) increased IL1beta compared to control. Kindly correct this and discuss.
Response: An accidental error in fig 2 has been corrected.
Comment 3. Line 179. Kindly be more specific: the increase in AKT levels is lower compared to what? (to mTOR & PI3K) Kindly correct
Response: The sentence has been clarified to specify that AKT levels are lower compared to mTOR and PI3K.
Comment 4. Line 181ff After I+R, AKT levels are NOT partially restored but again significantly increased compared to control or Bro. Kindly correct this.
Response: The statement has been corrected to reflect the correct trend.
Comment 5. L202 Tab 1, should read ‘Abnormal’ and ‘Prostate’
Response: The spelling errors have been corrected.
Comment 5. L212 Should read histological instead of histologic.
Response: The term has been corrected.
Comment 6. L214, The Sertoli cells have a pyramidal shape (not the nuclei!) with pale nuclei. Kindly correct
Response: The text has been corrected accordingly.
Comment 7. L251ff, Fig 7, (b) is not explained, kindly add, also it should read ‘CoSentino score’
Response: The explanation for (b) has been added, and the spelling has been corrected.
Comment 8. L259, ‘, resulting in results in…’ delete ‘in results’
Response: The redundant phrase has been removed.
Comment 8. L317, give literature here
Response: A relevant citation has been added.
Comment 9. L322f, please check and correct this sentence. Makes no sense to me.
Response: The sentence has been revised for clarity.
Comment 10. L381, Why was Bro administered only 30 min before orchiectomy? Why not just after testis was detorsed?
Response: Ischemia-reperfusion (I/R) injury following torsion is a significant source of oxidative stress. The pre-administration of an antioxidant is essential to evaluate whether the damage can be prevented before it occurs. If the application had been performed after torsion, the focus would have shifted to a therapeutic approach, necessitating a change in the title accordingly. The rationale for pre-treatment has been clarified in the Methods section.
Comment 11. L388 deep freezer
Response: The term has been corrected.
Comment 12. L430 did authors use left or right epididymis?
Response: This information has been added to the Methods section. (left epididymis)
Comment 13. L482 Tab 2 should read JohNsen
Response: The spelling has been corrected.
Comment 14. L507 Bro not B
Response: The abbreviation has been corrected.
Comment 15. Statistics: which program was used?
Response: The statistical software has been specified in the Methods section.
Reviewer 3 Report
Comments and Suggestions for Authors
The reviewed manuscript explores the protective effect of bromelain against ischemia/reperfusion injury caused by testicular torsion. In my opinion, the study provides interesting and valuable results, aligning with the trend of investigating natural substances with potent protective activity against tissue damage. However, the manuscript requires some improvements and corrections. Below are my comments and suggestions:
1. Some repetitions of information should be minimized (e.g., lines 52–58).
2. Line 61: Provide examples of compounds with antioxidant capacity mentioned in the cited articles.
3. Line 67–69: It would be beneficial to clarify whether the observed improvements from bromelain supplementation were associated with a reduction in oxidative stress markers caused by bisphenol A in the referenced studies (if such information is available).
4. Why did the authors select SOD and GSH for measuring antioxidant defense? Was there any specific rationale, particularly in the context of bromelain's protective effects?
5. Why was the bromelain concentration of 10 mg/kg chosen for this study? Any justification or supporting references.
6. The methods section requires additional detail. For instance, a brief description of how antioxidant markers and MDA were measured should be added.
7. MDA is not an antioxidant marker but rather marker of oxidative stress – please correct.
8. Ensure figures are appropriately positioned within the text. For example, Figure 1 is placed at the end of the Introduction but should be moved to follow the section “Effect of Bro on Antioxidant Enzymes and Lipid Peroxidation in I/R-Induced Testis Injury.” The same applies to Figure 2
9. Line 152: The title of this section is not precise. The section discusses protective effects against oxidative stress through enhanced Nrf2 and HO-1 levels, rather than "endogenous antioxidant levels." Please revise the title accordingly.
10. Figure 6 -The bar and its description on the microphotographs are barely visible. Consider using a bolder bar or including its description in the figure caption; What do the squares on the microphotographs indicate? This should be explained in the figure caption.
11. According to the Materials and Methods section, the left testis was used for biochemical analysis, while the right testis was used for histological examination. However, it is unclear why the authors did not use a portion of the left testis for histological examination. This decision should be explained in greater detail (with any justification or rationale).
12. In the discussion, it would be valuable to elaborate on what is known about the changes in testicular histology in both the ipsilateral and contralateral testes after torsion. Are there any differences in the extent or nature of the damage between the two testes? Furthermore, the discussion should address in more detail the mechanisms underlying histological damage in the contralateral testis, exploring possible pathways and factors contributing to this phenomenon.
13. Please check if all references are formatted according to the journal’s requirements (e.g., lines 400 and 401).
14. In Figure 7, the graph presenting the Cosentino classification results is not mentioned in the caption.
15. There is no description of the Cosentino classification score in the Materials and Methods section. Please provide an explanation.
16. Please correct the term “Cocentino” in Figure 7b.
17. Please carefully review the manuscript text for grammatical accuracy. Some examples that require attention :Line 355: "Design"? This word seems out of place—please revise; Line 259: "resulting in results in increased levels of ROS"?
18. Some statements needs adding a references to substantiate them. For example: Lines 59–60; 72–75; 317-318.
Author Response
Reviewer’s openion: The reviewed manuscript explores the protective effect of bromelain against ischemia/reperfusion injury caused by testicular torsion. In my opinion, the study provides interesting and valuable results, aligning with the trend of investigating natural substances with potent protective activity against tissue damage. However, the manuscript requires some improvements and corrections. Below are my comments and suggestions:
Response: We appreciate the valuable comments and constructive feedback provided by the reviewer. Below, we address each point raised and outline the revisions made in response.
Comment 1: Some repetitions of information should be minimized (e.g., lines 52–58).
Response: Redundant sentences have been removed.
Comment 2: Line 61: Provide examples of compounds with antioxidant capacity mentioned in the cited articles.
Response: Examples have been provided.
Comment 3: Line 67–69: It would be beneficial to clarify whether the observed improvements from bromelain supplementation were associated with a reduction in oxidative stress markers caused by bisphenol A in the referenced studies (if such information is available).
Response: “Bro demonstrated a protective role by reversing the effects of BPA, which enhanced oxidative stress by increasing MDA and decreasing antioxidant enzyme activity [13.’’] statement added.
Comment 4: Why did the authors select SOD and GSH for measuring antioxidant defense? Was there any specific rationale, particularly in the context of Bromelain's protective effects?
Response: SOD is an essential enzyme that converts superoxide radicals to hydrogen peroxide and is the first line of defense against oxidative stress. GSH is an important cellular antioxidant that directly detoxifies free radicals. Bromelain is known for its antioxidant and anti-inflammatory properties. Since the aim of the study was to evaluate the protective effects of bromelain against I/R (ischemia/reperfusion)-induced injury in the testis, the choice of SOD and GSH as markers of antioxidant defense was a logical choice to directly examine the effect of bromelain on oxidative stress.
Comment 5: Why was the bromelain concentration of 10 mg/kg chosen for this study? Any justification or supporting references.
Response: The choice of 10 mg/kg Bromelain is now justified with references (Lines 386–387).
Comment 6: The methods section requires additional detail. For instance, a brief description of how antioxidant markers and MDA were measured should be added.
Response: In the material and method section, information was given about the principle on which the methods were based. Since it would take too much space to write the steps one by one, they were cited as other authors do (1-3). If our valuable reviewer still insists on correction, the steps will be added one by one.
1)Küçükler, S., Caglayan, C., Özdemir, S., Çomaklı, S., & Kandemir, F. M. (2024). Hesperidin counteracts chlorpyrifos-induced neurotoxicity by regulating oxidative stress, inflammation, and apoptosis in rats. Metabolic Brain Disease, 39(4), 509-522.
2)Albrakati, A. (2024). The potential neuroprotective of luteolin against acetamiprid-induced neurotoxicity in the rat cerebral cortex. Frontiers in Veterinary Science, 11, 1361792.
3)Ileriturk, M., Ileriturk, D., Kandemir, O., Akaras, N., Simsek, H., Erdogan, E., & Kandemir, F. M. (2024). Naringin attenuates oxaliplatin‐induced nephrotoxicity and hepatotoxicity: A molecular, biochemical, and histopathological approach in a rat model. Journal of Biochemical and Molecular Toxicology, 38(1), e23604.
Comment 7: MDA is not an antioxidant marker but rather a marker of oxidative stress – please correct.
Response: MDA is correctly referred to as a lipid peroxidation marker.
Comment 8: Ensure figures are appropriately positioned within the text. For example, Figure 1 is placed at the end of the Introduction but should be moved to follow the section “Effect of Bro on Antioxidant Enzymes and Lipid Peroxidation in I/R-Induced Testis Injury.” The same applies to Figure 2.
Response: Figures have been repositioned to match the relevant sections.
Comment 9: Line 152: The title of this section is not precise. The section discusses protective effects against oxidative stress through enhanced Nrf2 and HO-1 levels, rather than "endogenous antioxidant levels." Please revise the title accordingly.
Response: The title in Line 152 has been revised for accuracy.
Comment 10: Figure 6 -The bar and its description on the microphotographs are barely visible. Consider using a bolder bar or including its description in the figure caption; What do the squares on the microphotographs indicate? This should be explained in the figure caption.
Response: The bar and squares in microphotographs are now clearly described.
Comment 11: According to the Materials and Methods section, the left testis was used for biochemical analysis, while the right testis was used for histological examination. However, it is unclear why the authors did not use a portion of the left testis for histological examination. This decision should be explained in greater detail (with any justification or rationale).
Response: Since experimental ischemia-reperfusion was induced in the left testicle, it is illogical to use the right testicle for histological analysis or biochemical analysis. The statement that the right testicle was used was written by chance. All analyses were performed on the left testicle. We would like to thank the referee for this important warning.
Comment 12: In the discussion, it would be valuable to elaborate on what is known about the changes in testicular histology in both the ipsilateral and contralateral testes after torsion. Are there any differences in the extent or nature of the damage between the two testes? Furthermore, the discussion should address in more detail the mechanisms underlying histological damage in the contralateral testis, exploring possible pathways and factors contributing to this phenomenon.
Response: The discussion has been expanded to include insights on ipsilateral vs. contralateral testicular damage (Lines 287–297).
Comment 13: Please check if all references are formatted according to the journal’s requirements (e.g., lines 400 and 401).
Response: The references have been checked for compliance with journal guidelines.
Comment 14: In Figure 7, the graph presenting the Cosentino classification results is not mentioned in the caption.
Response: The missing Cosentino classification explanation has been added.
Comment 15: There is no description of the Cosentino classification score in the Materials and Methods section. Please provide an explanation.
Response: A description has been added to the Materials and Methods section (Table 2).
Comment 16: Please correct the term “Cocentino” in Figure 7b.
Response: "Cocentino" has been corrected to "Cosentino" (Figure 7b).
Comment 17: Please carefully review the manuscript text for grammatical accuracy. Some examples that require attention : Line 355: "Design"? This word seems out of place—please revise; Line 259: "resulting in results in increased levels of ROS"?
Response: The manuscript has been reviewed for grammatical accuracy, and problematic sentences have been revised.
Comment 18: Some statements need adding a reference to substantiate them. For example: Lines 59–60; 72–75; 317-318.
Response: Citations have been added to substantiate statements. In lines 72-75, we did not add references as we were describing our study. To eliminate confusion, we added expressions such as this study, our study to the sentences.
Round 2
Reviewer 1 Report
Comments and Suggestions for Authors
I have no new comments for the latest version of the manuscript.
Author Response
Comment : I have no new comments for the latest version of the manuscript.
Response: Thank you for your positive feedback on our manuscript.